# Fuzzy-Modulus-Based Layered Elastic Analysis of Asphalt Pavements for Enhanced Fatigue Life Prediction

**DOI:** 10.3390/ma18133034

**Published:** 2025-06-26

**Authors:** Artur Zbiciak, Denys Volchok, Zofia Kozyra, Rafał Michalczyk, Nassir Al Garssi

**Affiliations:** 1Institute of Roads and Bridges, Faculty of Civil Engineering, Warsaw University of Technology, Al. Armii Ludowej 16, 00-637 Warsaw, Poland; artur.zbiciak@pw.edu.pl (A.Z.); abdraboh11@gmail.com (N.A.G.); 2Department of Structural and Theoretical Mechanics and Strength of Materials, Educational and Scientific Institute “Prydniprovska State Academy of Civil Engineering and Architecture”, Ukrainian State University of Science and Technology, 49010 Dnipro, Ukraine; denys.l.volchok@gmail.com; 3Institute of Building Engineering, Faculty of Civil Engineering, Warsaw University of Technology, Al. Armii Ludowej 16, 00-637 Warsaw, Poland; zofia.kozyra@pw.edu.pl

**Keywords:** flexible pavements, fatigue life prediction, fuzzy set theory, uncertainty modeling, mechanistic-empirical design

## Abstract

The paper presents a novel approach to evaluating the fatigue performance of asphalt pavements using fuzzy set theory to model the uncertainty in the elastic moduli of asphalt layers. The method integrates fuzzy numbers with an analytical multilayer elastic pavement model. By applying α-cut representation and defuzzification techniques, the model delivers fuzzy estimations of critical strain responses and associated fatigue lives under traffic loading. The proposed methodology captures uncertainty in material properties more realistically than conventional deterministic approaches. The effectiveness of this technique is demonstrated through the Asphalt Institute’s fatigue models for tensile and compressive strains. The results provide fuzzy bounds for fatigue life parameters and enable robust pavement design under material uncertainty. By incorporating fuzzy-modulus-based parameters into layered elastic half-space models, the proposed method significantly improves the predictive reliability of pavement performance.

## 1. Introduction

Increasing traffic volumes and escalating demands for pavement durability have led to substantial advancements in pavement design methods. Traditional empirical pavement design methods, exemplified by the widely used 1993 AASHTO Guide [1], rely on empirical relationships derived primarily from historical data and experimental observations. Although straightforward and historically effective [2,3], these methods have inherent limitations in reliably predicting pavement performance under the diverse and complex conditions encountered in practice, such as variable environmental conditions and evolving traffic demands. To overcome these limitations, mechanistic-empirical (M-E) approaches have been developed. These advanced methods explicitly integrate mechanical theories with empirical calibration [4], improving pavement performance prediction accuracy by accounting for traffic loading, material variability, and environmental influences [5,6].

A cornerstone of mechanistic pavement design is the layered elastic theory introduced by Burmister [7], which continues to provide foundational solutions for stresses and displacements in multilayered pavement structures subjected to loads. The enduring relevance of Burmister’s solutions lies in their rigorous theoretical foundation and adaptability to various pavement conditions. The layered elastic theory utilized by the Asphalt Institute is among the most established mechanistic frameworks for asphalt pavement design and applies classical multi-layered elastic models to predict pavement responses and subsequent fatigue and rutting performance [8]. Although robust, the deterministic assumption of, e.g., constant elastic moduli, fails to reflect the inherent variability in asphalt mixtures. The variability in pavement behavior arises notably from temperature fluctuations, material aging, and inherent construction variability. For instance, Brown et al. [9] investigated the performance of hot mix asphalt (HMA), highlighting how variations in mix design and compaction practices directly impact pavement durability. Roberts et al. [10] extensively documented the importance of precise HMA mixture designs and how deviations can significantly affect pavement response and longevity. Additionally, Kandhal and Mallick [11] examined the critical influence of recycled materials and their variable properties on pavement performance, emphasizing the complexities arising from recycled asphalt pavements (RAPs) that contribute to material property uncertainties. This challenge of material uncertainty is mirrored in the operational phase of pavement life, where data-driven approaches such as machine learning are becoming essential for the non-destructive detection and classification of the resulting defects [12,13]. Meanwhile, the Mechanistic-Empirical Pavement Design Guide (MEPDG) [14] systematically integrates mechanical principles with empirical data, recognizing variability in pavement responses driven by climatic conditions, loading patterns, and material properties, thus offering improved predictive capabilities compared to purely empirical approaches.

Addressing uncertainty in pavement analysis requires methods capable of capturing epistemic uncertainties (those resulting from incomplete or imprecise information). Fuzzy set theory, initially proposed by Zadeh [15], provides an effective mathematical framework for modeling such uncertainties, particularly valuable when comprehensive statistical data is unavailable or impractical to obtain. Unlike deterministic numbers, which represent precise and exact values, fuzzy numbers are used to express uncertainty, imprecision, or ambiguity in data. Instead of assigning a single fixed value, fuzzy numbers characterize quantities through ranges associated with varying degrees of confidence or membership. Each fuzzy number is described by a membership function, typically ranging from zero (no membership) to one (full membership), reflecting how strongly each value within the range belongs to the fuzzy set. This approach allows for a more realistic representation of uncertain or approximate information which cannot be accurately captured by single numerical values alone. Recent applications of fuzzy logic in pavement engineering have shown promise in improving prediction accuracy and decision-making robustness [16,17,18,19]. Fuzzy logic is particularly valuable in pavement engineering due to its inherent capability to handle uncertainty arising from imprecise, vague, or insufficiently defined input parameters. Beyond modeling uncertainties related solely to the modulus of elasticity, fuzzy methods have been successfully applied to other critical pavement parameters such as variations in material strength and environmental influences, including temperature fluctuations, aging phenomena, and variability introduced by construction processes [20,21]. By effectively capturing these diverse sources of uncertainty, fuzzy logic enhances the robustness and reliability of pavement performance predictions, enabling more informed and flexible engineering decisions

Burmister’s solution, foundational in layered elastic analysis, involves evaluating infinite integrals with Bessel functions, presenting computational challenges. Additionally, application of the α-cut-based layered elastic model inherently involves increased computational overhead compared to traditional deterministic approaches. Specifically, the necessity of repeatedly solving the numerical integration problems for multiple α-levels significantly extends calculation times. To mitigate this issue, efficient numerical integration techniques, such as adaptive quadrature or Gauss–Laguerre methods, have been employed, enhancing computational efficiency without sacrificing accuracy [22,23,24,25]. Additionally, parallel computation strategies and code optimization within MATLAB 2024b were implemented to reduce overall processing times, ensuring the method remains practical for engineering applications despite its increased computational demands.

This paper integrates fuzzy set theory with layered elastic theory for pavement analysis, aiming at more realistic predictions of fatigue life by explicitly accounting for uncertainties in pavement materials. The approach is validated through MATLAB’s own code simulations, highlighting significant improvements over deterministic analyses. A comprehensive numerical example demonstrates how fuzzy logic enhances critical strain estimations compared to classical deterministic approaches. Results indicate that fuzzy analysis not only captures realistic pavement behavior more accurately but also provides engineers with a robust tool for optimizing design strategies to extend pavement service life.

## 2. Materials and Methods

### 2.1. Asphalt Pavement Design Methodologies

Pavement design methodologies have evolved significantly over the past decades, moving from purely empirical approaches towards advanced mechanistic-empirical (M-E) frameworks. Each methodology is characterized by distinct assumptions, analytical rigor, and empirical calibration, influencing their applicability and accuracy in pavement design practice.

The Asphalt Institute (AI) method is among the most widely applied mechanistic-empirical approaches, utilizing layered elastic theory to estimate pavement responses, such as tensile strain at the bottom of asphalt layers and vertical compressive strain on subgrade layers. These parameters predict fatigue cracking and rutting performance [5,7,8]. The AI method, however, relies on deterministic inputs, limiting its ability to represent the variability inherent in pavement materials and construction processes.

The mechanistic-empirical methodology developed by the Asphalt Institute offers a well-established framework for the structural design of asphalt pavement systems. This approach is based on the evaluation of mechanical responses—primarily tensile and compressive strains—within the pavement structure under repetitive traffic loading. These responses are then used as inputs to empirical fatigue and rutting models, calibrated from laboratory and field data.

The fatigue life $N_{f}$, expressed as the number of standard axle repetitions to the initiation of bottom-up cracking, is calculated using a semi-empirical model:(1)$$N_{f}=18.4\cdot C\cdot \left(\;6.167\cdot 10^{-5}\cdot {\varepsilon_{H}}^{-3.291}\cdot E^{-\;0.854}\right)$$
where $\varepsilon_{H}$ is the horizontal tensile strain at the bottom of the asphalt layer, E [MPa] is the modulus of the asphalt mixture, and C is a mixture-specific correction factor defined as:(2)$$C=10^{M},\;M=4.84\cdot \left(\frac{V_{a}}{V_{a}+V_{v}}-0.69\right)$$
with $V_{a}\;[\%]$ and $V_{v}\;[\%]$ denoting the air void and total void contents of the mixture, respectively. This relationship reflects the influence of volumetric parameters on the fatigue resistance of asphalt concrete.

Rutting life $N_{r}$, associated with permanent deformation in the subgrade, is estimated using a strain-based model:(3)$$N_{r}={\left(\frac{k}{\left|\varepsilon_{V}\right|}\right)}^{1/m}$$
where $\varepsilon_{V}$ is the vertical compressive strain at the top of the subgrade, and k and m are empirical coefficients derived from calibration studies.

Accurate evaluation of $\varepsilon_{H}$ and $\varepsilon_{V}$ is critical to this methodology (see Figure 1). These strain values depend on the material properties and geometry of the pavement system, as well as the characteristics of the applied load. To compute them, a rigorous analytical–numerical procedure was implemented based on classical elasticity theory for axisymmetric layered media.

The AASHTO (American Association of State Highway and Transportation Officials) design methodologies have significantly influenced pavement engineering practice. The empirical AASHTO Guide is based on road tests conducted in the late 1950s and early 1960s, offering straightforward procedures but limited accuracy due to outdated empirical data [1,2,3]. To overcome these limitations, the Mechanistic-Empirical Pavement Design Guide (MEPDG), developed under NCHRP Project 1-37A, integrates mechanical models with empirical relationships to predict pavement distress more accurately, accounting explicitly for climatic, material, and loading conditions [4,6,14].

The Shell pavement design method developed by Shell Laboratories offers another prominent mechanistic-empirical approach. This method emphasizes comprehensive modeling of asphalt mixture behavior, predicting pavement performance through empirical fatigue and rutting criteria based on extensive laboratory tests and field observations [26,27,28].

The French LCPC method (Laboratoire Central des Ponts et Chaussées) employs layered elastic theory coupled with empirical distress models calibrated from long-term performance data. This approach distinguishes itself by explicitly considering viscoelastic and plastic behavior of asphalt materials, offering enhanced accuracy for conditions typical in Europe [29,30,31].

Additional methods, such as the Austroads Guide and the British TRRL (Transport and Road Research Laboratory) method, reflect localized adaptations of mechanistic-empirical principles tailored to specific climates, traffic conditions, and construction practices in Australia and the UK respectively, each contributing unique insights and empirical calibrations to global pavement design practice [9,32,33].

Selecting the appropriate pavement design method depends on project-specific conditions, data availability, desired accuracy, and regional empirical calibrations. Modern trends advocate combining multiple approaches or employing probabilistic analyses to account for uncertainties and enhance the robustness of pavement designs.

### 2.2. Layered Elastic Half-Space Analysis

Layered elastic half-space analysis provides a rigorous method to determine stress and strain distributions within pavement systems. This approach is crucial for accurate predictions of pavement performance, particularly fatigue and rutting life [2,8].

The pavement is modeled as a set of homogeneous, linearly elastic layers perfectly bonded to each other and resting on a semi-infinite elastic half-space [7,34,35]. The system is loaded with a uniform circular pressure distribution p, with contact radius a, representing a single wheel load. Owing to the axisymmetric nature of the problem, the governing equations of elasticity are formulated in cylindrical coordinates and reduced to a scalar biharmonic equation.

In each layer, the displacement field is derived from a stress potential function $\Phi_{i}\left(r,z\right)$ written in the form:(4)$$\Phi_{i}\left(r,z\right)=\int_{0}^{\infty }\left(A_{\;i}\;e^{-\eta \;z}+B_{i}\;z\;e^{-\eta \;z}+C_{\;i}\;e^{\;\eta \;z}+D_{\;i}\;z\;e^{\;\eta \;z}\right)J_{0}\left(\eta r\right)\;d\eta \equiv \int_{0}^{\infty }{\hat{\Phi }}_{i}\left(r,z\right)\;d\eta$$
where $J_{0}$ is the Bessel function of the first kind and order zero, and η is the radial wave number introduced via the Hankel transform. For the half-space (subgrade), only decaying terms are used to ensure boundedness at infinity:(5)$$\Phi_{\infty }\left(r,z\right)=\int_{0}^{\infty }\left(E_{\;\infty }\;e^{-\eta \;z}+F_{\infty }\;z\;e^{-\eta \;z}\right)J_{0}\left(\eta r\right)\;d\eta \equiv \int_{0}^{\infty }{\hat{\Phi }}_{\infty }\left(r,z\right)\;d\eta$$

The stress, strain, and displacement fields in each layer are computed by differentiating Φ according to the classical relations. For example, selected stress and displacement component are as follows:(6)$$\sigma_{zz}=\frac{\partial }{\partial \;z}\left[\left(2-\nu \right)\nabla^{2}\Phi -\frac{\partial^{2}\Phi }{\partial \;z^{2}}\right],\;\sigma_{rz}=\frac{\partial }{\partial \;r}\left[\left(1-\nu \right)\nabla^{2}\Phi -\frac{\partial^{2}\Phi }{\partial \;z^{2}}\right]$$(7)$$u_{z}=\frac{1}{2\;\mu }\left[2\left(1-\nu \right)\nabla^{2}\Phi -\frac{\partial^{2}\Phi }{\partial \;z^{2}}\right],\;u_{\;r}=-\frac{1}{2\;\mu }\frac{\partial^{2}\Phi }{\partial \;r\;\partial \;z},\;\mathrm{where}\;\mu =\frac{E}{2\left(1+\nu \right)}$$

Here, $\nabla^{2}\Phi$ is the Laplacian of the potential function, ν is Poisson’s ratio, and μ is the shear modulus. The boundary conditions at the surface and the continuity conditions at each interface yield a linear system of equations for the unknown coefficients $A_{i}$,$B_{i}$,$C_{i}$,$D_{i}$, which are functions of η.

The surface pressure is transformed into the Bessel domain as:(8)$$p\left(r\right)=q\;a\;\int_{0}^{\infty }J_{0}\left(\eta r\right)J_{1}\left(\eta a\right)\;d\eta$$
where $J_{1}$ is the Bessel function of the first kind and order one. The infinite integrals containing Bessel functions and exponential terms are evaluated numerically for each $\eta \in [\left.0,\infty \right)$. The final strain components at the desired depth and radial position are obtained by integrating the evaluated expressions over η, using numerical quadrature:(9)$$\varepsilon_{H}=\int_{0}^{\infty }f_{H}\left(\eta \right)\;d\eta \;,\;\varepsilon_{V}=\int_{0}^{\infty }f_{V}\left(\eta \right)\;d\eta$$
where $f_{H}\left(\eta \right)$ and $f_{V}\left(\eta \right)$ represent the horizontal and vertical strain components expressed as functions of the integration variable and material parameters.

The evaluation of infinite integrals containing Bessel functions is computationally intensive. This challenge is magnified in a fuzzy analysis, where the entire mechanical model must be solved repeatedly for numerous α-cuts to propagate uncertainty. Therefore, the focus of the mechanical implementation was on optimizing this numerical procedure to ensure both computational accuracy and efficiency.

The layered elastic analysis presented herein was implemented independently in two computational environments, Maple 2024 and MATLAB, combining symbolic algebra to construct the governing expressions with numerical solvers and integration routines. The model allows for flexible definition of the pavement structure, including varying number of layers, material stiffness, and thicknesses. The core computational steps and logic of the analysis are detailed in a pseudocode available in the Appendix A.

A comparative evaluation of the numerical results obtained from both programs demonstrated good agreement, confirming the accuracy and reliability of the developed numerical codes.

Figure 2 illustrates the contour plots of horizontal and vertical strains, respectively, for a typical asphalt pavement structure, which will be analyzed in detail in Section 3. The pavement configuration corresponds to the data summarized in Table 1 (presented in Section 3), assuming modal values for the stiffness moduli of asphalt layers. These deterministic results provide a reference scenario for further fuzzy-based analysis.

Numerical results indicate that this method provides accurate strain evaluations suitable for mechanistic-empirical pavement design. The derived values of $\varepsilon_{H}$ and $\varepsilon_{V}$ serve directly as input to fatigue and rutting life models (see Equations (4) and (6)) and reflect the influence of both structural configuration and material composition.

This approach offers a powerful hybrid method for pavement analysis, bridging classical elasticity theory with modern computational tools. It is particularly well-suited for advanced design scenarios requiring detailed evaluation of stress–strain behavior in complex, multilayered systems.

### 2.3. Application of Fuzzy Set Theory in Structural Mechanics

The application of fuzzy set theory [15] in structural mechanics has become increasingly significant [36,37,38,39], particularly in the context of pavement engineering. In pavement structures, uncertainties related to material properties, loading conditions, and environmental influences pose challenges for accurate fatigue life predictions. Fuzzy-modulus-based layered elastic analysis, addressed in the current study, demonstrates the efficacy of fuzzy logic in handling these uncertainties to enhance the reliability and accuracy of fatigue life predictions for asphalt pavements.

One of the prominent applications of fuzzy set theory in pavement mechanics is modeling the variability and uncertainty in material properties, such as the modulus of elasticity. Conventional deterministic approaches are limited in their ability to accurately represent real-world variability. Employing fuzzy modulus values enables the incorporation of uncertainty and spatial variability of asphalt concrete properties, significantly improving fatigue life prediction accuracy [40,41].

Integrating fuzzy logic into reliability assessments allows more nuanced characterization of pavement performance. Fuzzy reliability methods incorporate qualitative assessments and expert opinions, complementing traditional probabilistic methods [38,42,43]. This combined approach facilitates more realistic modeling of structural reliability, particularly under uncertain traffic loads and environmental conditions.

The fuzzy logic framework is effectively applied to damage identification and pavement condition assessments, where uncertainties and subjective judgments are prevalent. Fuzzy inference systems (FISs) can systematically interpret ambiguous inspection data, translating them into actionable insights regarding pavement maintenance strategies [44,45].

Fuzzy set theory significantly contributes to the optimization and decision-making processes involved in pavement design and management. By accommodating multiple conflicting objectives and uncertainties, fuzzy optimization methods provide robust and cost-effective solutions that balance performance, cost, and reliability [36,37,39].

In the context of the fuzzy-modulus-based layered elastic analysis of asphalt pavements, the application of fuzzy set theory offers a robust methodological approach that addresses critical uncertainties. Through improved representation of material properties, enhanced reliability analysis, and effective optimization methods, fuzzy approaches significantly enhance fatigue life prediction accuracy and pavement structural integrity.

## 3. Results and Discussion

The numerical example demonstrates the application of fuzzy-modulus-based layered elastic analysis to evaluate and enhance fatigue life predictions of asphalt pavements. The example is solved numerically using a code programmed in MATLAB, employing assumptions that closely align with realistic pavement conditions [2,7,8].

The numerical analysis presented in this study is designed to estimate the fatigue and rutting life of flexible asphalt pavement structures using a mechanistic-empirical approach enhanced with fuzzy logic to incorporate material uncertainty [15,41,46]. The pavement system is modeled as a five-layer elastic medium subjected to axisymmetric loading conditions [7,34]. The governing equations are derived from classical elasticity theory and solved analytically using stress functions and Fourier–Bessel series expansion [8,22,23,24].

The analyzed pavement configuration consists of the following layers (from top to bottom): wearing course, binder course, asphalt base, treated subbase, and natural subgrade. The mechanical behavior of each layer is characterized by linear elasticity, with layer-specific Young’s moduli $E_{i}$ and a constant Poisson’s ratio ν=0.3 applied throughout.

The surface loading condition represents a circular tire footprint with a uniform pressure of q=650 kPa and a contact radius a=0.1565 m, simulating a single-wheel load. The load is modeled as a Bessel-type function in the radial domain to reflect the axisymmetric distribution of stresses beneath the wheel.

Material stiffness values for the three asphalt layers (wearing, binder, and base) are defined using fuzzy numbers to reflect inherent variability in production and construction processes, as well as temperature operating conditions. Each modulus is described as a non-classical fuzzy number with five characteristic points: minimum, lower semi-core, mode, upper semi-core, and maximum values (see Figure 2a). For instance, the wearing course modulus spans from 2500 MPa to 20,000 MPa, with a modal value of 10,300 MPa (see Table 1, and compare Figure 3).

The five-point membership functions were determined based on typical elastic moduli values recommended by widely used pavement design catalogs and guidelines complemented by expert judgment and laboratory validation. The minimum and maximum values were derived considering extreme operational scenarios reflecting significant variations in asphalt modulus due to seasonal temperature fluctuations, as well as potential material aging effects. Such ranges, along with the modal values reflecting typical recommended moduli, are explicitly presented in established national and international design guidelines (e.g., AASHTO recommendations, Asphalt Institute Manuals). Finally, the intermediate semi-core points represent realistic statistical confidence intervals determined based on variability reported in pavement condition surveys and laboratory data sets routinely cited in pavement engineering literature [1,5,14].

The analysis proceeds by evaluating pavement responses over a sequence of α-cuts (with 51 discrete levels), representing confidence intervals within the fuzzy sets. At each α-level, interval arithmetic [47] is applied to generate lower and upper bounds of horizontal tensile strain ($\varepsilon_{H}$) at the bottom of the asphalt base and vertical compressive strain ($\varepsilon_{V}$) at the top of the subgrade. The application of fuzzy arithmetic, specifically, in solving the local optimization problem for each alpha level is due to the large computational requirements of the considered problem [36,37,47].

We will refer to α as the level (or α—cut) of the fuzzy set A⊆X; such a crisp set $A_{\alpha }$ for which the condition $\mu_{A}\geq \alpha$ holds $\alpha \in \left[0,1\right]$ for $\alpha \in \left[0,1\right]$:(10)$$A_{\;\alpha }=\left\{\;x\in X:\;\mu_{A}(x)\geq \alpha \;\right\}$$A set $A_{\alpha }$ is defined by its characteristic function $\chi_{A\alpha }(x)$:(11)$$\chi_{A\alpha }(x)=\left\{\begin{array}{cc} 1, & \mu_{A}(x)\geq \alpha \\ 0, & \mu_{A}(x)<\alpha \end{array}\right.$$

The graphical results provide a comprehensive view of how uncertainty in asphalt material properties influences the mechanistic response of pavement structures and, consequently, their predicted fatigue and rutting performance. The fuzzy approach adopted in this study allows for a full-field representation of variability, moving beyond conventional deterministic or probabilistic methods.

Figure 3 presents the normalized fuzzy membership functions of the Young’s moduli using Equation (12), for the wearing, binder, and base courses, respectively. It is important to emphasize that in the fuzzy methodology presented, the shape and spread of these functions serve as the direct graphical representation of uncertainty. This approach fulfills a role analogous to that of statistical error bars but within a non-probabilistic framework consistent with fuzzy set theory. All three layers are modeled using a non-classical linearized convex membership function derived from the data [40,41,48], with clear modal values and asymmetric spreads on either side.(12)$$\mu_{A}^{N}(x)=\frac{\mu_{A}(x)}{h(A)}$$
where $h=\underset{x\in A}{\max}\mu_{A}(x)$ is the maximum value of a function.

These normalized distributions reflect the range of plausible stiffnesses encountered in real-world conditions due to production, aging, and temperature conditions. The functions can be further improved based on even more detailed information about weather conditions. The wearing course exhibits the broadest uncertainty range, while the base course shows a relatively narrower core but a longer tail, indicating sensitivity to extreme scenarios. These fuzzy input parameters form the basis for the propagation of uncertainty throughout the mechanistic model.

The next operation in fuzzy modeling is the decomposition operation, which represents a set A⊆X in the form of a collection of crisp sets $A_{\alpha }$:(13)$$A=\underset{\alpha \in \;\left[0,1\right]}{\cup }\;\alpha A_{\;\alpha }$$
where $\alpha A_{\alpha }$ defines a fuzzy set, where elements are assigned corresponding degrees of membership, that is:(14)$$\mu_{A_{\;\alpha }}=\left\{\begin{array}{cc} \mu , & x\in A_{\;\alpha }\\ 0, & x\notin A_{\;\alpha } \end{array}\right.$$

The reverse statement is also true. If there is a set of subsets of a certain universal set X and a set of numbers from $\left[0,1\right]$, then there is a synthesis of a fuzzy set in X in accordance with Equations (13) and (14). This is an important property of fuzzy set theory, as it allows forming fuzzy sets of multiple results using the extension principle. Let us consider it in more detail.

Let f be a crisp bijective mapping of the space X into the space Y; that is:(15)f:   X→Y

Let A fuzzy set be defined in X:(16)$$A=\sum_{j=1}^{n}\frac{\mu_{A}(x_{i})}{x_{i}}\;,\;A\subseteq X$$

Let the set B be generated by a mapping f and defined in the space Y. Then, the following holds:(17)$$B=f(A)=\sum_{i=1}^{n}\frac{\mu_{A}(x_{i})}{f(x_{i})}$$
or(18)$$B=f(A)=\sum_{i=1}^{n}\frac{\mu_{B}(y_{i})}{y_{i}},\;y_{i}=f(x_{i}),\;x_{i}\in X$$
in which(19)$$\mu_{B}\left(y\right)=\left\{\begin{array}{ll} \underset{x\in f^{\;-1}\left(y\right)}{\sup}\;\mu_{A}\left(x\right), & m_{x}\in f^{\;-1}\left(y\right)\\ \mu_{A}\left(x\right), & f^{\;-1}\left(y\right)=\left\{x\right\}\\ 0, & f^{\;-1}\left(y\right)=\emptyset \end{array}\right.$$
where ∅ denotes the empty set of elements, and $f^{-1}$ is the inverse transformation y=f(x); that is, $f^{-1}:\;Y\to X$, $m_{x}$- the modal value of x.

Equation (18) represents the so-called *generalization principle* (or extension). It allows transferring (extending) various mathematical operations from crisp sets to fuzzy sets.

The resulting horizontal tensile strain ($\varepsilon_{H}$) at the bottom of the asphalt base and the vertical compressive strain ($\varepsilon_{V}$) at the top of the subgrade are shown in Figure 4 and Figure 5. Both are expressed as fuzzy numbers derived from interval evaluations across 51 α-cut levels. The strain envelopes exhibit a characteristic narrowing with increasing α, capturing how uncertainty diminishes as confidence in the input grows. Notably, the tensile strain $\varepsilon_{H}$ displays a wider uncertainty band from 25.73 × 10^−6^ to 97.31 × 10^−6^ than $\varepsilon_{V}$, ranging from −269.37 × 10^−6^ to −91.96 × 10^−6^, underscoring its greater sensitivity to asphalt stiffness variability. This has direct implications for fatigue design, where small shifts in tensile strain can lead to orders-of-magnitude differences in predicted life.

These differences are explicitly visible in Figure 6, which illustrates the fuzzy fatigue life $N_{f}$. The fuzzy set spans multiple decades on the horizontal axis, revealing that uncertainty in layer stiffness alone can yield a variation in fatigue life of more than one order of magnitude. The modal fatigue life lies roughly at the center of this range, but the envelope highlights the risk of both overly conservative and overly optimistic predictions if a single point estimate is used. A similar pattern emerges in Figure 7 for the rutting life $N_{r}$.

To obtain usable scalar outputs for design decisions, several defuzzification methods were applied to both $N_{f}$ and $N_{r}$ (see Figure 8 and Figure 9 respectively). Among them, the centroid and bisector methods yielded values that are lower than the modal values (see Table 2), suggesting that the central tendency of the fuzzy set lies toward the conservative side. This outcome is particularly relevant in the context of safety-critical infrastructure, where underestimation of damage potential is more acceptable than overestimation.

$\mathrm{Fatigue}\;\mathrm{life}\;N_{f}$$\mathrm{Rutting}\;\mathrm{life}\;N_{r}$ Based on the results presented in Table 2, it can be observed that fatigue and rutting life predictions obtained using the centroid defuzzification method (29.55 × 10^6^ and 736.57 × 10^6^, respectively) are slightly higher compared to those from the bisector method (28.30 × 10^6^ and 689.24 × 10^6^, respectively). The differences, although modest, highlight how the choice of defuzzification method can influence final pavement life estimations. Therefore, the selection between the centroid and bisector methods should be guided by the engineer’s expert judgment and a practical balance between construction costs and desired pavement durability.

The analysis reveals that reliance on modal values alone—as is common in conventional mechanistic-empirical design—can obscure the true variability in pavement response. The fuzzy approach not only exposes this hidden uncertainty but also quantifies its potential impact on design life. In particular, the fatigue criterion is shown to be highly sensitive to input stiffness distributions, reinforcing the need for robust design frameworks that account for material uncertainty beyond simple factor-of-safety adjustments.

The fuzzy modulus approach effectively captures uncertainties associated with pavement material properties. Modulus variability presented in the results emphasizes how fuzzy analysis can realistically simulate in-service pavement conditions, improving fatigue life predictions by accounting for uncertainties inherent in pavement materials.

This study demonstrates the integration of fuzzy logic with mechanistic-empirical pavement design to address the epistemic uncertainty inherent in asphalt material properties. By modeling the Young’s moduli of asphalt layers as fuzzy numbers and propagating these uncertainties through an analytical multilayer elastic solution, the resulting pavement responses and performance indicators were expressed as fuzzy sets.

## 4. Conclusions

The conducted research highlights the substantial advantages of integrating fuzzy set theory with traditional layered elastic pavement analysis methods. Unlike conventional deterministic models, the fuzzy-modulus-based approach systematically accounts for uncertainties in material characteristics, environmental influences, and loading variations. The MATLAB-based numerical analysis clearly illustrates that the fuzzy model provides a more realistic estimation of pavement responses, particularly critical strains associated with fatigue failures. sets.

The key findings can be summarized as follows:Uncertainty in asphalt stiffness parameters significantly affects critical strain responses, particularly horizontal tensile strain at the base of the asphalt layer, which, in turn, influences fatigue life predictions.The use of fuzzy logic revealed wide ranges in predicted fatigue and rutting life, indicating that deterministic or single-point estimates may misrepresent the true performance of the pavement structure.Defuzzification methods such as the centroid and bisector approaches provided more realistic scalar estimates of pavement life, accounting for the full distribution of possible outcomes.

The proposed framework offers a robust alternative to conventional design by incorporating uncertainty directly into the computational process rather than addressing it through ad hoc safety factors. This method provides greater transparency in the decision-making process and supports the development of more resilient and reliable pavement structures.

Key findings from the numerical example demonstrate that deterministic analyses, although useful, inherently assume precise input parameters. However, real pavement structures exhibit significant variations due to natural heterogeneity of materials, construction inaccuracies, and unpredictable environmental factors. Fuzzy analysis effectively addresses these uncertainties, generating a range of plausible pavement responses rather than a single, deterministic solution. Consequently, this provides designers with valuable insights into the sensitivity of pavement performance relative to input uncertainties, enhancing decision-making processes.

Moreover, fuzzy-modulus-based analysis offers practical benefits in pavement management and maintenance strategies. The improved accuracy in predicting fatigue life can significantly optimize maintenance schedules, reduce costs, and enhance the long-term reliability of asphalt pavements. Such an approach not only facilitates more informed design decisions but also encourages proactive rather than reactive maintenance practices.

It is important, however, to qualify the context of these advantages in relation to other uncertainty quantification methods. The presented approach provides a valuable alternative primarily to deterministic methods by effectively capturing epistemic uncertainties, which are common in pavement engineering when precise statistical data is limited or unavailable. We acknowledge that when comprehensive statistical data are abundant, established probabilistic approaches are more appropriate and precise. Thus, the method presented here is not intended to supplant probabilistic techniques in data-rich scenarios but to offer a robust framework for decision-making under the conditions of uncertainty frequently encountered in practice.

In conclusion, the integration of fuzzy logic into pavement design methodologies represents a substantial advancement beyond traditional deterministic methods. It provides engineers with a powerful analytical framework capable of addressing real-world complexities, ultimately leading to enhanced pavement durability, reduced life-cycle costs, and improved safety standards. Future research should focus on further refining fuzzy models and extending their applications to various pavement configurations and materials, thereby broadening the potential for widespread adoption within pavement engineering practice.

## Figures and Tables

**Figure 1 materials-18-03034-f001:**
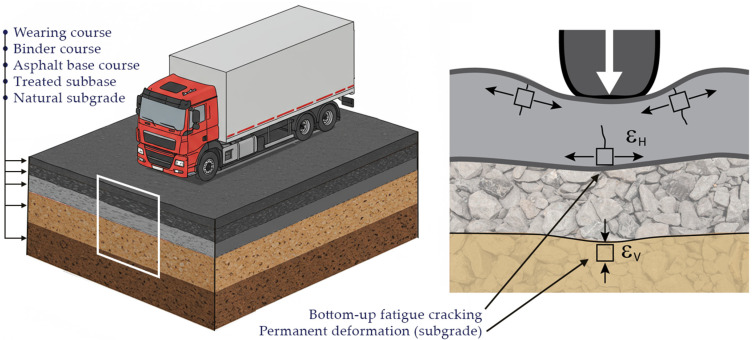
Flexible pavement (layers labeled) illustrating bottom-up fatigue cracking in asphalt (tensile strain) and subgrade permanent deformation (compressive strain).

**Figure 2 materials-18-03034-f002:**
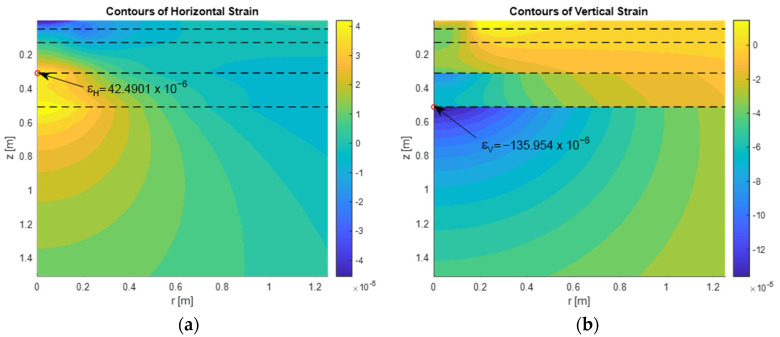
(**a**) Contour plots of horizontal strain. (**b**) Contour plots of vertical strain.

**Figure 3 materials-18-03034-f003:**
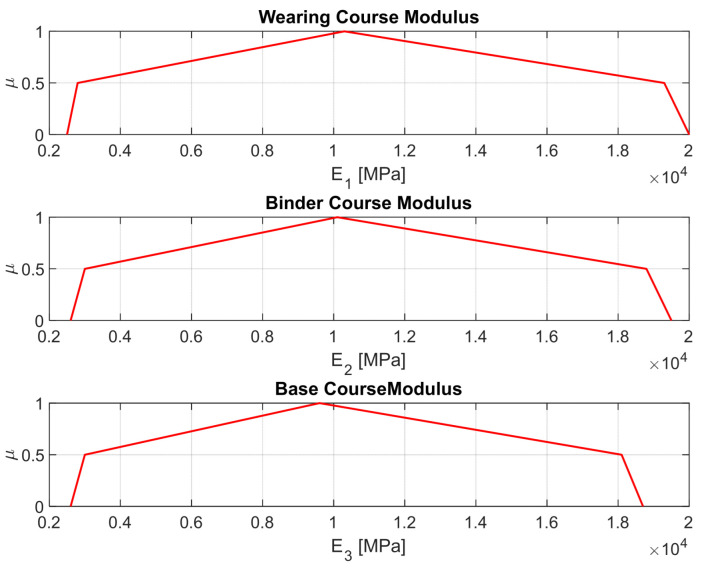
Asphalt pavement layers as fuzzy numbers.

**Figure 4 materials-18-03034-f004:**
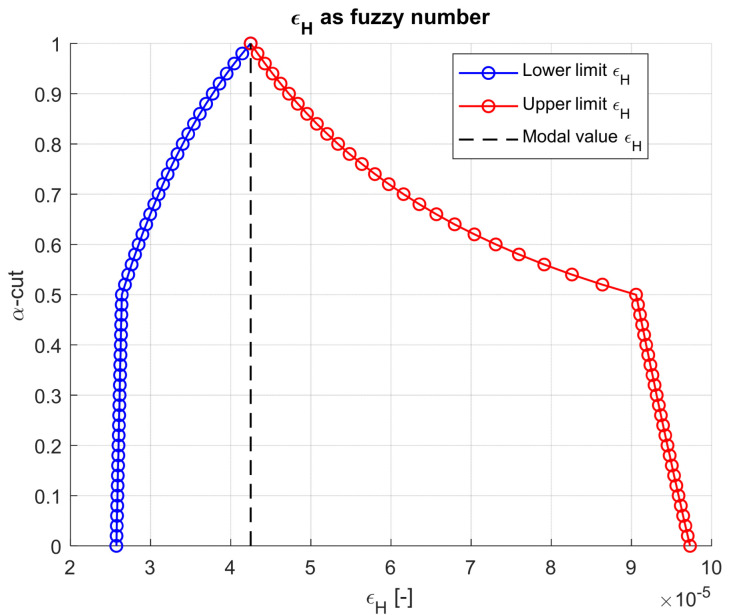
Horizontal strain at the bottom of asphaltic layers as fuzzy result.

**Figure 5 materials-18-03034-f005:**
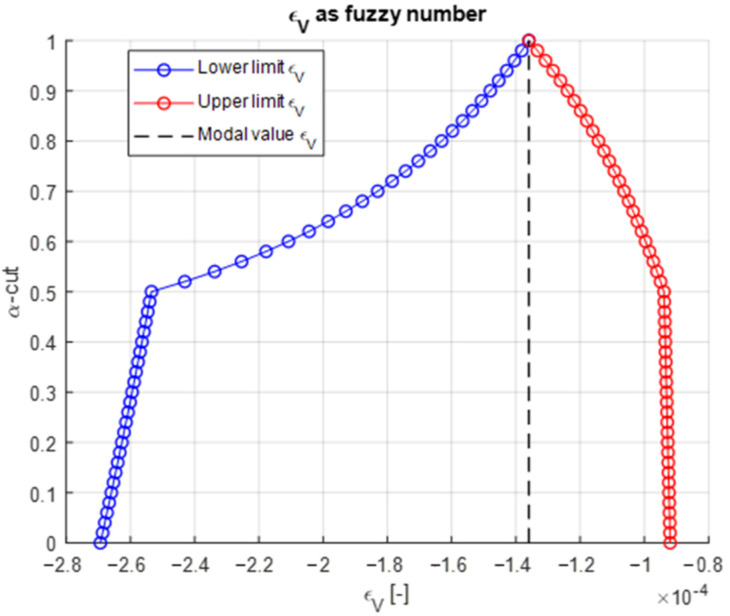
Vertical strain at the top of soil base as fuzzy result.

**Figure 6 materials-18-03034-f006:**
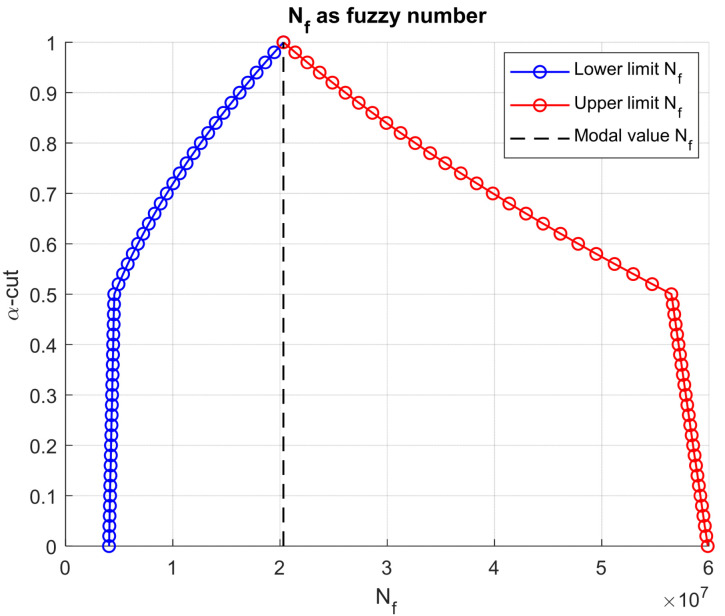
Resulting fatigue life as fuzzy number.

**Figure 7 materials-18-03034-f007:**
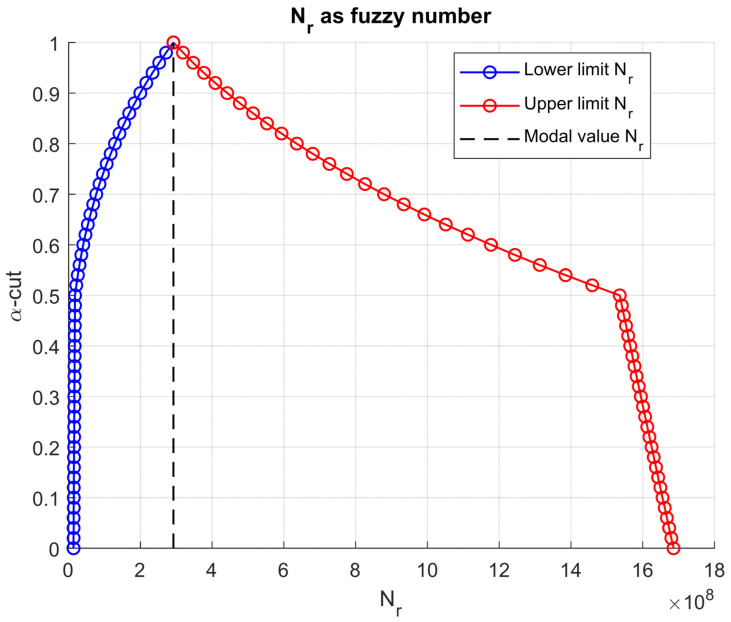
Resulting rutting life as fuzzy number.

**Figure 8 materials-18-03034-f008:**
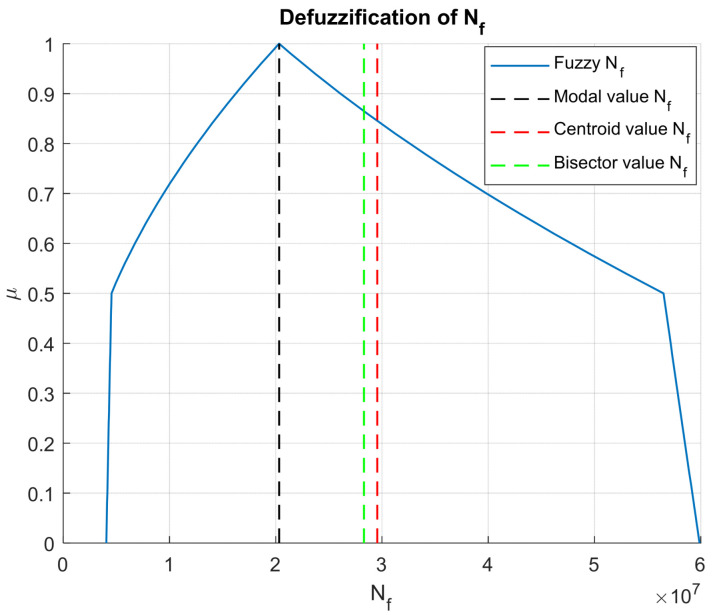
Fatigue life defuzzification results.

**Figure 9 materials-18-03034-f009:**
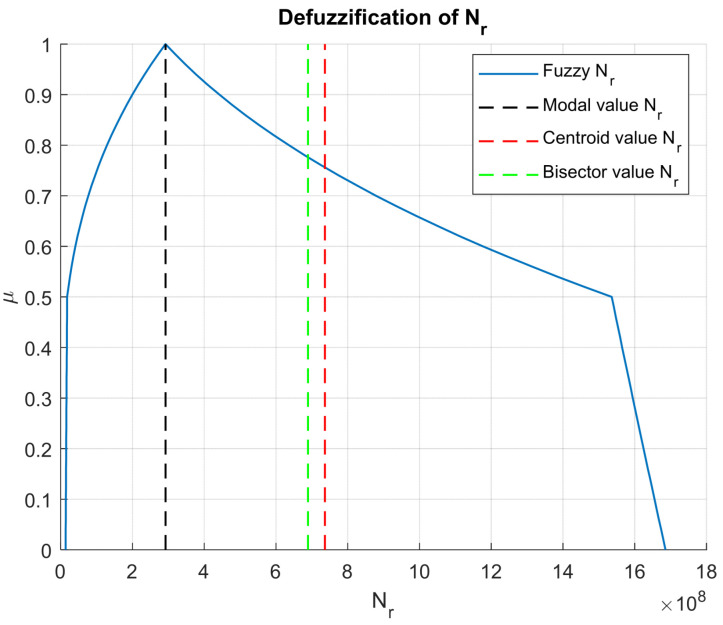
Rutting life defuzzification results.

**Table 1 materials-18-03034-t001:** Input parameters for numerical analysis.

Layer	Type	Thickness [m]	Elastic Modulus (Modal Value) [MPa]	Poisson’s Ratio [-]	Fuzzy Definition
Wearing course	Asphalt	0.05	2500–20,000 (10,300)	0.3	Non-classical
Binder course	Asphalt	0.08	2600–19,500 (10,100)	0.3	Non-classical
Asphalt base course	Asphalt	0.18	2600–18,700 (9600)	0.3	Non-classical
Treated subbase	Aggregate	0.20	400	0.3	Deterministic
Natural subgrade	Soil	∞ (semi-infinite)	100	0.3	Deterministic

**Table 2 materials-18-03034-t002:** Summary of fatigue and rutting life results.

Life Type	Min Value	Modal Value	Max Value	Centroid	Bisector
$\mathrm{Fatigue}\;\mathrm{life}\;N_{f}$	4.07 × 10^6^	20.32 × 10^6^	59.89 × 10^6^	29.55 × 10^6^	28.30 × 10^6^
$\mathrm{Rutting}\;\mathrm{life}\;N_{r}$	13.61 × 10^6^	292.05 × 10^6^	1685.80 × 10^6^	736.57 × 10^6^	689.24 × 10^6^

## Data Availability

The original contributions presented in this study are included in the article/Appendix A. Further inquiries can be directed to the corresponding author.

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
