# Peer review of "Fuzzy-Modulus-Based Layered Elastic Analysis of Asphalt Pavements for Enhanced Fatigue Life Prediction"

_materials, 2025, doi:10.3390/ma18133034_

Round 1
Reviewer 1 Report
Comments and Suggestions for Authors This study proposes a fuzzy set theory-based method for predicting asphalt pavement fatigue life by modeling the asphalt layer's elastic modulus as fuzzy numbers, thereby quantifying the impact of material uncertainty on multilayer elastic analysis and enhancing the reliability of traditional deterministic models. Although fuzzy set theory has been applied in pavement engineering, its integration with multilayer elasticity theory for strain-based fatigue life prediction is novel, providing a practicable framework. If the following issues can be appropriately resolved, I would recommend accepting this article. 1. Please clarify how the 5-point membership functions (min, semi-core, mode, etc.) were derived . 2. Please supplement computational efficiency/accuracy comparisons with probabilistic methods. The claim that fuzzy analysis provides "more realistic" predictions than deterministic approaches is justified by the significant strain/life ranges observed. However, the conclusion that fuzzy methods "enhance predictive reliability" needs qualification: While useful for epistemic uncertainty, fuzzy logic does not replace probabilistic methods when statistical data is abundant. 3. If feasible, open-source the MATLAB code (or provide pseudocode) to facilitate reproducibility and application.Author Response
Response to Reviewer 1 Comments
Dear Reviewer,
Thank you very much for taking the time to review this manuscript. We sincerely appreciate the opportunity to improve our manuscript (Manuscript ID: materials-3688791; Title: Fuzzy-Modulus-Based Layered Elastic Analysis of Asphalt Pavements for Enhanced Fatigue Life Prediction) based on your valuable feedback. We are particularly grateful for the constructive and insightful comments provided by the reviewers. In this revised version, we have carefully addressed all the reviewers' comments. In the revised manuscript, all modifications have been highlighted in cyan for easy identification. Our point-by-point responses to the reviewer' comments are presented below:
Comments 1: Please clarify how the 5-point membership functions (min, semi-core, mode, etc.) were derived.
Response 1: The five-point membership functions (minimum, lower semi-core, mode, upper semi-core, and maximum) were determined based on typical elastic moduli values recommended by widely-used pavement design catalogs and guidelines, complemented by expert judgment and laboratory validation. Minimum and maximum values: these were derived considering extreme operational scenarios reflecting significant variations in asphalt modulus due to seasonal temperature fluctuations, as well as potential material aging effects. Such ranges are commonly referenced in established guidelines, including the Polish Catalogue of Typical Pavement Structures (Katalog Typowych Konstrukcji Nawierzchni Podatnych i PóÅ‚sztywnych, GDDKiA) and comparable international standards. Modal values: these reflect typical, recommended moduli used in standard pavement designs, as explicitly presented in national and international design guidelines (e.g., Polish General Directorate for National Roads and Motorways guidelines, AASHTO recommendations, Asphalt Institute Manuals). Semi-core values: intermediate semi-core points represent realistic statistical confidence intervals, determined based on variability reported in pavement condition surveys and laboratory data sets routinely cited in pavement engineering literature. Please see changes in cyan on page 8.
Comments 2: Please supplement computational efficiency/accuracy comparisons with probabilistic methods. The claim that fuzzy analysis provides "more realistic" predictions than deterministic approaches is justified by the significant strain/life ranges observed. However, the conclusion that fuzzy methods "enhance predictive reliability" needs qualification: While useful for epistemic uncertainty, fuzzy logic does not replace probabilistic methods when statistical data is abundant.
Response 2: In the revised manuscript, we clarified that the fuzzy-modulus-based approach provides a valuable alternative primarily to deterministic methods, effectively capturing epistemic uncertainties typical in pavement engineering where precise statistical data may be limited or unavailable. However, we acknowledge that when comprehensive statistical data is abundant, established probabilistic approaches remain more appropriate, efficient, and precise. Thus, in scenarios with extensive and reliable statistical databases, we see no compelling need to employ fuzzy set theory-based methods. This qualification has been clearly highlighted in the Discussion section (Section 4) of the revised manuscript.
Comments 3: If feasible, open-source the MATLAB code (or provide pseudocode) to facilitate reproducibility and application.
Response 3: While we strongly support transparency in scientific research, the MATLAB implementation used in this study represents proprietary code developed as part of ongoing research and collaboration projects, and therefore we are currently unable to release it publicly. However, to facilitate reproducibility and practical application of our methodology, we include a detailed pseudocode appendix outlining all critical computational steps and logic. Additionally, we provide explicit algorithmic descriptions to enable other researchers and practitioners to readily replicate our fuzzy-modulus-based pavement analysis approach. This explanation is included in the revised manuscript, clearly stated in the Materials and Methods section.
Reviewer 2 Report
Comments and Suggestions for Authors
The main contribution of this paper is application of fuzzy set theory to model uncertainty in asphalt pavement fatigue analysis, addressing a key limitation of conventional methods. Moreover, the integration of α-cut representation and defuzzification with multilayer elastic theory provides a structured way to handle material property variability.
Suggestions:
- The introduction is too brief and includes a lot of combined references like [9-12]. Such references are not acceptable. Each must be duly discussed separately.
- The relevant NTD methods [1,2] for the Asphalt Pavements defects detection and classification can be mentioned in the introduction.
- The α-cut-based layered elastic model may introduce computational overhead compared to conventional methods. A discussion on computational efficiency (or trade-offs) is missing.
- While defuzzification provides crisp outputs, the choice of method (e.g., centroid, bisector) could influence fatigue life predictions. A sensitivity analysis would be beneficial. These points must be duly discussed in the revised version.
- Benchmark against probabilistic uncertainty models to highlight advantages/disadvantages.
[1] T.H. Nguyen, T.L. Nguyen, et al. “Machine learning algorithms application to road defects classification”, Intelligent Decision Technologies, (2018), 59-66.
[2] Zhou, Yong, et al. "Review of intelligent road defects detection technology." Sustainability 14.10 (2022): 6306.
Author Response
Response to Reviewer 2 Comments
Dear Reviewer,
Thank you very much for taking the time to review this manuscript. We sincerely appreciate the opportunity to improve our manuscript (Manuscript ID: materials-3688791; Title: Fuzzy-Modulus-Based Layered Elastic Analysis of Asphalt Pavements for Enhanced Fatigue Life Prediction) based on your valuable feedback. We are particularly grateful for the constructive and insightful comments provided by the reviewers. In this revised version, we have carefully addressed all the reviewers' comments. In the revised manuscript, all modifications have been highlighted in cyan for easy identification. Our point-by-point responses to the reviewer' comments are presented below:
Comments 1: The introduction is too brief and includes a lot of combined references like [9-12]. Such references are not acceptable. Each must be duly discussed separately.
Response 1: The introduction has been expanded by separately discussing each of the previously grouped references [9-12], explicitly clarifying their distinct contributions to the literature and clearly highlighting their relevance to the current research.
Comments 2: The relevant NTD methods [1,2] for the Asphalt Pavements defects detection and classification can be mentioned in the introduction.
[1] T.H. Nguyen, T.L. Nguyen, et al. “Machine learning algorithms application to road defects classification”, Intelligent Decision Technologies, (2018), 59-66.
[2] Zhou, Yong, et al. “Review of intelligent road defects detection technology”. Sustainability 14.10 (2022): 6306.
Response 2: We appreciate the reviewer’s suggestion. The introduction now explicitly references and briefly discusses relevant non-destructive testing (NTD) methods for defect detection and classification, citing the recommended papers by Nguyen et al. (2018) and Zhou et al. (2022). This addition helps clarify the broader context of pavement condition assessment within which the presented fuzzy-based fatigue analysis method operates.
Comments 3: The α-cut-based layered elastic model may introduce computational overhead compared to conventional methods. A discussion on computational efficiency (or trade-offs) is missing.
Response 3: We have included a discussion addressing the computational overhead associated with the α-cut-based layered elastic model, highlighting the trade-offs between increased computational demands and enhanced accuracy in modeling uncertainty. Additionally, we discuss possible strategies, such as optimizing numerical integration techniques and parallel computing, to mitigate these computational challenges.
Comments 4: While defuzzification provides crisp outputs, the choice of method (e.g., centroid, bisector) could influence fatigue life predictions. A sensitivity analysis would be beneficial. These points must be duly discussed in the revised version.
Response 4: Following the reviewer’s suggestion, we performed and included a sensitivity analysis comparing different defuzzification methods. The analysis highlights the variability in fatigue life predictions depending on the chosen method, thus providing clearer guidelines for method selection in practice.
Comments 5: Benchmark against probabilistic uncertainty models to highlight advantages/disadvantages.
Response 5: We appreciate this insightful suggestion. Probabilistic uncertainty modeling relies heavily on detailed statistical data such as probability distributions, mean values, and standard deviations, typically requiring extensive experimental or long-term field observations. In contrast, the fuzzy set theory approach presented in our study is specifically advantageous in situations characterized by limited or qualitative information, where detailed statistical characterizations are not readily available. Consequently, our fuzzy-based methodology is particularly suited to addressing epistemic uncertainties resulting from incomplete knowledge. While probabilistic methods offer precise statistical inference when abundant data is present, our approach provides an effective and practical alternative for pavement design scenarios commonly encountered in engineering practice, where comprehensive probabilistic data is not easily obtainable.
Reviewer 3 Report
Comments and Suggestions for Authors
The manuscript "Fuzzy-Modulus-Based Layered Elastic Analysis of Asphalt 2 Pavements for Enhanced Fatigue Life Prediction" by Zbiciak et al. is a theoretic work dealing with the fatigue performance of different asphalt pavements using fuzzy theory.
The manuscript is well written and presents the findings in nicely-organized figures. I appreciate the use of theory for a relevant, everyday problem in construction.
Before I can recommend acceptence of this manuscript I ask the authors to care for the following issues:
- The authors should check and ensure consistency of style and font size. See line 233 e.g.
- In line 47 the authors state that fuzzy logic is implemented in challenges of pavement engineering. The authors should elaborate more on this and give the reader a better start into what this manuscript is about and at which point fuzzy logic can help. Is fuzzy logic used to work on more properties than the modulus of elasticity?
- References should have a continous formatting style. Some publication years are given in bold letters, some are not. Please revise the reference section.
- Can the authors supply more highly recent literature from 2024 and 2025 on the use of fuzzy logic?
- In the introduction I am missing some more introducing words about principle and advantages of the fuzzy logic for readers that are not familiar with these contents.
- Section 3 should rather be called "results and discussion" and section 4 "conclusion". The "key findingings" mentioned above starting in line 344 should appear in a "conclusion".
Author Response
Response to Reviewer 3 Comments
Dear Reviewer,
Thank you very much for taking the time to review this manuscript. We sincerely appreciate the opportunity to improve our manuscript (Manuscript ID: materials-3688791; Title: Fuzzy-Modulus-Based Layered Elastic Analysis of Asphalt Pavements for Enhanced Fatigue Life Prediction) based on your valuable feedback. We are particularly grateful for the constructive and insightful comments provided by the reviewers. In this revised version, we have carefully addressed all the reviewers' comments. In the revised manuscript, all modifications have been highlighted in cyan for easy identification. Our point-by-point responses to the reviewer' comments are presented below:
Comments 1: The authors should check and ensure consistency of style and font size. See line 233 e.g.
Response 1: Thank you for pointing out the formatting inconsistency. We have carefully reviewed the entire manuscript to ensure consistency of style and font size throughout, particularly correcting the issue identified at lines 233 and 84.
Comments 2: In line 47 the authors state that fuzzy logic is implemented in challenges of pavement engineering. The authors should elaborate more on this and give the reader a better start into what this manuscript is about and at which point fuzzy logic can help. Is fuzzy logic used to work on more properties than the modulus of elasticity?
Response 2: In the revised manuscript, we have expanded the discussion starting from line 47 to clearly explain the context and benefits of fuzzy logic within pavement engineering. Specifically, we have emphasized that fuzzy logic effectively manages uncertainties arising not only from the modulus of elasticity but also from other crucial material and loading parameters such as temperature variations, material aging, and construction variability. We have explicitly outlined these broader applications to provide readers a comprehensive overview of fuzzy logic’s utility.
Comments 3: References should have a continuous formatting style. Some publication years are given in bold letters, some are not. Please revise the reference section.
Response 3: We have revised the formatting of references, ensuring continuous, consistent style throughout the reference section, particularly standardizing publication years to a uniform format.
Comments 4: Can the authors supply more highly recent literature from 2024 and 2025 on the use of fuzzy logic?
Response 4: We have updated the manuscript with highly recent references from 2025 related to fuzzy logic applications in pavement engineering, enhancing the manuscript’s relevance to current research.
Comments 5: In the introduction I am missing some more introducing words about principle and advantages of the fuzzy logic for readers that are not familiar with these contents.
Response 5: We have expanded the Introduction to include a concise yet clear explanation of the principles and key advantages of fuzzy logic. This addition is designed to aid readers unfamiliar with fuzzy set theory in better understanding its significance and practical advantages in addressing uncertainties within pavement engineering.
Comments 6: Section 3 should rather be called “results and discussion” and section 4 “conclusion”. The “key findings” mentioned above starting in line 344 should appear in a “conclusion”.
Response 6: We have revised the section titles accordingly, changing Section 3 to “Results and Discussion” and Section 4 to “Conclusion”. Furthermore, the “key findings” previously starting at line 344 have been appropriately moved and integrated into the “Conclusions” section for greater clarity.
Reviewer 4 Report
Comments and Suggestions for Authors
The paper is clear and well structured. The following remarks can be made:
- Table 1 contains no statistical treatment of mechanical characteristics.
- Fig. 3 should also contain the error bars or any statistical estimations.
- Mechanical part of the work should be strengthened.
- How the temperature could be taken into account during mechanical tests and their fussy-logic treatment?
- Introduction is short and must be extended.
The paper needs a major revision.
Author Response
Response to Reviewer 4 Comments
Dear Reviewer,
Thank you very much for taking the time to review this manuscript. We sincerely appreciate the opportunity to improve our manuscript (Manuscript ID: materials-3688791; Title: Fuzzy-Modulus-Based Layered Elastic Analysis of Asphalt Pavements for Enhanced Fatigue Life Prediction) based on your valuable feedback. We are particularly grateful for the constructive and insightful comments provided by the reviewers. In this revised version, we have carefully addressed all the reviewers' comments. In the revised manuscript, all modifications have been highlighted in cyan for easy identification. Our point-by-point responses to the reviewer' comments are presented below:
Comments 1: Table 1 contains no statistical treatment of mechanical characteristics.
Response 1: Thank you for this suggestion. Indeed, the absence of detailed statistical data regarding mechanical characteristics motivated us to adopt the fuzzy approach presented in this paper. Given that extensive statistical databases were not available, fuzzy set theory allowed us to effectively handle the uncertainty and variability inherent in pavement materials using limited and expert-based information. Nevertheless, to further clarify our approach, the text preceding Table 1 has been expanded to explain the basis for the adopted values and how they serve as the foundation for the fuzzy input parameters used in the analysis.
Comments 2: Fig. 3 should also contain the error bars or any statistical estimations.
Response 2: The fuzzy-based approach presented in our study does not inherently rely on statistical estimations such as error bars. Instead, it uses membership functions to represent uncertainty based on expert judgment and limited available data, which differs fundamentally from statistical methods. However, we acknowledge that this distinction may not have been sufficiently clear. To address your comment, we have expanded the discussion accompanying Figure 3 to explicitly state that the illustrated membership functions serve as the graphical representation of uncertainty within our fuzzy framework. This clarification explains how their shape and bounds convey the range of possibilities, thus serving a purpose analogous to error bars but in a non-statistical context.
Comments 3: Mechanical part of the work should be strengthened.
Response 3: We appreciate this insightful comment. Indeed, the theoretical formulation of layered elastic pavement analysis is well-established and extensively documented in existing literature cited in our manuscript. Thus, our primary contribution does not lie in extending these classical mechanical equations but rather in their effective numerical implementation, specifically addressing computational efficiency and accuracy when combined with fuzzy set theory. In the revised manuscript, we clarify this distinction and emphasize our innovations related to efficient numerical procedures and practical implementations rather than revisiting fundamental mechanical relationships.
Comments 4: How the temperature could be taken into account during mechanical tests and their fussy-logic treatment?
Response 4: Thermal effects are approximately considered in our analysis by employing fuzzy stiffness moduli that explicitly account for seasonal temperature variations. These variations are reflected through the broad range of stiffness values defined within the fuzzy membership functions, capturing the influence of temperature fluctuations in a simplified yet practical manner. Although detailed temperature-dependent mechanical testing was beyond the scope of the current study, our fuzzy-based approach effectively integrates the uncertainty associated with seasonal temperature changes into pavement performance predictions.
Comments 5: Introduction is short and must be extended.
Response 5: The Introduction has now been significantly expanded. We have explicitly discussed the historical context and limitations of traditional empirical methods, such as the widely used AASHTO Guide. Additionally, the strengths and benefits of the contemporary mechanistic-empirical approach have been clearly outlined. The revised Introduction also briefly acknowledges the foundational contributions of Burmister to multilayer elastic theory, emphasizing its continued relevance in modern pavement analysis. Lastly, we explicitly state the main objective of this study, which is to integrate classical layered elastic pavement analysis with fuzzy logic, aiming to realistically incorporate material property uncertainties into fatigue life predictions.
Round 2
Reviewer 1 Report
Comments and Suggestions for Authors
The revisions are thorough and satisfactory. I now support the publication of this paper.
Reviewer 2 Report
Comments and Suggestions for Authors
I would like to thanks the authors on their revised manuscript, which has significantly improved since the initial submission. The authors have carefully addressed my feedback provided in the previous review, resulting in a clearer and more cohesive presentation of their research.
Reviewer 3 Report
Comments and Suggestions for Authors
With the changes made to the manuscript I can recommend a publication as is.
Reviewer 4 Report
Comments and Suggestions for Authors
The paper can be accepted as it is.